# Evaluation of Athletes’ Food Choices during Competition with Use of Digital Images

**DOI:** 10.3390/nu11071627

**Published:** 2019-07-17

**Authors:** Fiona E. Pelly, Rachael Thurecht

**Affiliations:** School of Health and Sport Sciences, University of the Sunshine Coast, Maroochydore, QLD 4558, Australia

**Keywords:** food choice, food selection, athletes, competition, diet quality, digital images, digital photography

## Abstract

The selection of foods made by athletes during competition can impact performance, yet to date, the quality of their food choices has not been investigated. The aim of this study was to describe the food selection of athletes in a buffet-style dining hall setting in terms of diet quality, food variety, and volume of food and compare to their self-rating of their meal, reasons for the choosing the food items, access to previous nutrition advice, and use of nutrition labelling. A total of 81 athletes (42 females, 39 males) from 24 sports across 58 countries at the 2018 Commonwealth Games (Qld, Australia) participated in this study. A digital photograph was taken of the athletes’ meal after selection from the buffet and prior to consumption. Each participant was asked a series of questions in relation to their food selection. The photographs were coded into recommended serves of food groups based on the Australian Guide to Healthy Eating. The nutritional analysis and photograph of a standard serve size were used to quantify the energy and nutrients for the meal. Most athletes chose adequate quantities of macronutrients, which agreed with their reasons for the food choice, but the majority did not include fruit (80.2%) or dairy (65.4%) in their food selection, while 54% of males included discretionary foods (0.25–7.0 serves). The median self-rating for food choice was 8/10. Most reasons for food choices were nutritional attributes, sensory factors, performance, usual eating practices and physiological factors (e.g., satiety, gut comfort). This suggests that athletes may need more education on the quality of food selected from buffet settings.

## 1. Introduction

There is currently a move towards food-based research away from a focus on nutrients in the general population [1], but this is less evident in athletes where current recommendations focus on the intake and timing of macronutrients [2]. While this is of interest to researchers and practitioners, in practice, athletes often lack the skills to translate nutrient recommendations into appropriate food choices. Recent studies have shown that athletes may not meet general guidelines in terms of serves of food groups and variety of food [3], which may ultimately impact general health as well as performance. The diet quality of athletes has been reported in a small number of studies as reviewed by Capling et al. [4] but is often not the primary focus of the research and has not been assessed broadly across different cohorts of athletes. It is evident that nutrition knowledge does not necessarily translate into appropriate dietary intake [5], and further research is needed to determine dietary adequacy of athletes [6].

Athletes are unique in that they are often exposed to many and varied food environments that are likely to influence food selection and subsequently the quality of their diet. Athletes travel to competition and training venues and commonly live in an unfamiliar environment with a different food supply. Live-in buffet-style dining halls present many options for the athlete, resulting in difficult choices about what to eat. The sheer volume of offerings can be confusing and potentially lead to over- or underconsumption or inappropriate food choices, impacting dietary intake, as has been previously demonstrated in this environment [3]. Established determinants of food choice, such as sensory appeal or influence of others [7,8], may play more of a role in the food that is selected.

Despite environmental influences being a key determinant in food choice [7], little is known about the influence the food environment has on food selection for athletes from a large-scale dining facility. Most studies have focused on individual factors, such as the athlete’s perception of the food itself [9] or the use of nutrition labelling [10], with a handful of studies looking at self-reported determinants of food choice of athletes [11,12,13,14], and only two focused on the food choice within this setting [15,16]. To date, no studies have looked at the actual food choices that athletes make in the dining hall environment. This is important to consider as the food environment can have an influence on dietary intake as athletes are constrained by the food choices on offer, which may influence their dietary intake and subsequent performance.

Use of photography provides a novel way to visually assess the food selection of athletes during and post-competition. Previous studies have used digital images to assess real-world environments, such as self-serve dining settings, and have been shown to provide objective and reliable measures of food selection and servings [17,18]. To date, image-assisted assessment of dietary intake has only been trialed as a method to assess dietary intake of athletes [19] and has not been used to observe their food choices, particularly in relation to the quality of their diet.

The aim of this study was to describe the food selection of athletes in a buffet-style dining hall setting in terms of diet quality, food variety and volume of food and compare to their self-rating of their meal, reasons for the choosing the food items, access to previous nutrition advice and use of nutrition labelling.

## 2. Materials and Methods

### 2.1. Setting

This research took place during the 2018 Commonwealth Games (Qld, Australia) within the athletes’ village dining hall in April 2018. The dining hall consisted of a range of service areas that provided both hot and cold menu items. Each item on the menu was coded to Australian Food Composition database (NUTTAB, 2010) and analysed by the research team in advance of the event using custom-made nutritional analysis software (CaterNut (www.caternut.com); Version 1.0, Maroochydore, Australia). During the event, each item was plated on a standard dining plate as a single serve on three separate occasions, and was weighed, photographed and labelled. The mean weight was determined to be the standard serve.

### 2.2. Participants

Athletes who were present in the main dining hall of the athletes’ village were invited to participate in this study. Researchers approached athletes returning to their seat immediately after they made their entire food selection for their meal. Each individual was provided with information about the study prior to giving their informed consent for inclusion. The study was conducted in accordance with the Declaration of Helsinki, and the protocol was approved by The University of the Sunshine Coast Human Research Ethics Committee (No. S1/71/086).

### 2.3. Data Collection

A photograph was taken of the meal using a tablet at an approximate 45-degree angle with a fiducial marker. All food was served into standard sized disposable plates and bowls, and serving trays that were provided in this environment. With the exception of juice, plain milk, tea and coffee, all drinks were bottled and labelled. The participants were asked a series of short questions that were entered into SurveyMonkey by the researcher. The questions included demographic characteristics, source of nutrition information, self-reported influences on their food selection, any dietary regimen followed, competition phase and self-rating of their food selection on a scale of 1 (very poor) to 10 (excellent) framed in the context of performance nutrition. Open-ended responses to individual reasons for food choice were preferred over existing survey tools as the purpose of the study was to investigate specific reasons for the meal selection, not broader factors influencing food choice. Field notes were taken by the researcher in the situation where the image did not accurately show all items on the tray. This study primarily aimed to investigate food choice, not actual dietary intake; therefore, waste was not recorded.

### 2.4. Data Analysis

Each image was assigned a unique identifier that was linked to the participant survey and was reviewed by two researchers to identify the menu items on the plate. This was complemented by the photographs of standard serves and field notes that recorded additional information about the food selection. Both researchers cross-referenced their food selection to ensure consistency.

The coding system used for food groups was adapted from Christoph et al. [18]. Some modifications were made to this method to suit the Australian context and food environment. A total of three coders all experienced in dietary analysis undertook the coding. All coders were provided with both training and a coding framework by the lead researcher. Visual examination of the image of the meal and researcher notes, along with the images and dietary analysis of standard serves and review of ingredient lists of menu items, were used to determine the portions of grains, fruit, vegetables (starchy and non-starchy), dairy, meats, legumes and discretionary food groups within each meal to the nearest ¼ cup. A minimum of 0.25 servings was required for the food group to be present. The Australian Guide to Healthy Eating (AGHE) [20] recommended serve sizes (e.g., ½ cup cooked vegetables = 1 serve) were used to estimate the number of serves of the food group for each item. Total fruit was separated from fruit juice, starchy vegetables (potato, corn and peas) were separated from other vegetables and legumes were classified separately. Discretionary foods (i.e., food items that are recommended to be consumed in limited quantities, such as confectionary or processed meats) were classified according to the AGHE and based one serve equivalent to 600 kJ. The nutrition analysis and ingredient list of each menu item was used to determine the number of serves in each discretionary item. The meal photographs were compared to the standard serves to determine any variation in food groups from mixed dishes (for example, the amount of beef in a lasagna slice may vary dependent on the amount of topping versus filling in the serve). Food groups were coded as being present or absent, and as the number of serves of each item. The total serves of core food groups (all groups minus discretionary items), the variety (number of different items on the plate) and the volume (number of plates of food) were also calculated. As per Christoph et al. [18], the lead coder was considered the “standard” that was compared to other coders for checking reliability and consistency. The interrater agreement to the lead coder was assessed randomly on 20 meals (25% of the sample). The average agreement between coders was 82%, with all items being greater than 90% with the exception of serves of starchy vegetables (72%), non-starchy vegetables (72%), grains (70%) and serves discretionary items (70%). All coders were in 100% agreement on the presence of foods on plates with the exception of discretionary items and starchy vegetables (80 and 85%, respectively).

Energy, protein, carbohydrate, fat, sodium and fibre content of each menu item in the photograph was based on the estimated serve size. This was also determined by visually comparing the meal image of the item to the image of the standard serve of the same item. In addition, each researcher had access to a reference plate and bowl, the fiducial marker and the field notes. The prior nutritional analysis of each menu item was used to quantify the energy and nutrients for the meal serve. The estimated quantity of the nutrient for the meal as a whole was determined by adding all values together, plus any additional items that were listed in the field notes but not visible in the image.

The results from both the qualitative and quantitative dietary assessment and survey were combined and entered into Statistic Packages for Social Science (SPSS) statistical analysis software (Version 24.0, IBM Corporation). Athletes were grouped according to sport (endurance, power/print, racket, team, skill- and weight-focused) and region (Africa, Australia/New Zealand, British Isles, Canada, Caribbean and Asia/Pacific) for analysis. As no variables were normally distributed, nonparametric tests (Mann–Whitney U test (MW-U), Kruskal–Wallis (KW) ANOVA with Bonferroni correction, Chi-square test and Spearman’s correlation) were used to compare differences between groups and relationships between variables. As the nature of the study was not designed to accurately determine the exact quantity of each nutrient in the meal, energy and nutrients were described as a median (Md) score with an interquartile range (IQR) and relevant food groups represented as a range. In addition, quantitative analysis of energy (kJ), protein (g), carbohydrate (g), fat (g), fibre (g) and sodium (mg) were grouped into categories of negligible, low, medium and high. Categories were determined by the researchers based on consideration of Australian food labelling standards [21], nutrient reference values [22], the AGHE [20] and sports nutrition consensus statements [2]. Dietary avoidance or regimens were coded into four categories based on performance factors, personal preferences, allergy/intolerance and macronutrient manipulation. Open-ended responses on reasons for food selection were coded and themed by the authors as per the method by Kondraki et al. [23].

## 3. Results

### 3.1. Participant Characteristics

A total of 81 athletes (42 females, 39 males) from 24 sports across 58 countries participated in this study. The demographic characteristics are shown in Table 1. Those that had previously experienced the dining hall environment were more likely to have had previous nutrition advice (Chi-square 4.954; *p* = 0.026) and were on average significantly older (*p* = 0.027).

### 3.2. Analysis of Meal Selection

More meals were tested prior to and during the individual’s competition (69.1%) than after completion (30.9%). There was no significant difference between the proportions of meals tested before/during and after competition across the different meal periods breakfast, lunch and dinner period (Table 2), nor were there any differences in the sex distribution across meals.

The estimated quantitative and qualitative analysis of the meals is presented in Table 3. Males selected greater volume of food and meals higher in energy, carbohydrate, sodium and fibre. Males were also more likely to make discretionary food choices and choose more serves of grains/cereals. There were also significant differences across meal periods with fewer choices of foods containing protein, grains, non-starchy vegetables and meats at breakfast in comparison to other meals (Table 3).

Figure 1 shows the sex differences for the quantitative meal assessment grouped into negligible, low, medium and high ranges for each nutrient. The majority of athlete meals contained >1700 kJ, 20–50 g protein, 10–40 g fat, 30–100 g carbohydrate, 5–10 g fibre and >400 mg sodium. There was a significant difference between sexes for energy (kJ) and carbohydrate (g), with a higher proportion of males classified as very high (>2800 kJ and >100 g) than females (Figure 1).

As expected, there were a number of correlations between nutrients and food groups in the meals. Only those with a significance of *p* < 0.01 have been reported. There was a positive relationship between energy (kJ) and serves of grains (*r* = 0.54, *p* < 0.001), meats (*r* = 0.33, *p* = 0.002), core food groups (*r* = 0.46, *p* < 0.001) and discretionary items (*r* = 0.46, *p* < 0.001); protein (g) and serves of meat (*r* = 0.62, *p* < 0.001), discretionary items (*r* = 0.46, *p* < 0.001) and core food groups (*r* = 0.24, *p* = 0.002); carbohydrate (g) and grains (*r* = 0.62, *p* < 0.001), serves of fruit (*r* = 0.45, *p* < 0.001), fruit juice (*r* = 0.37, *p* = 0.001) and discretionary items (*r* = 0.34, *p* = 0.002); and fat (g) with serves of meats (*r* = 0.056, *p* < 0.001), grains (*r* = 0.31, *p* = 0.004) and discretionary items (*r* = 0.40, *p* < 0.001). Sodium (mg) was correlated with serves of discretionary items (*r* = 0.56, *p* < 0.001) and meats (*r* = 0.56, *p* < 0.001), while fibre (g) was correlated to serves of grains (*r* = 0.40, *p* < 0.001).

The median grams of fibre (MW-U 505.5; *p* = 0.046) was significantly more in meals selected prior and during (Md 8.0 g) competition than post (Md 5.0 g). There were no apparent differences between different regions and nutrient composition of the meals.

Athletes’ self-rating of their meal selection was a median (range) of 8 (2–10) out of 10. There was a positive correlation between age and self-rating of food selection (*r* = 0.276, *p* = 0.013), with younger athletes rating their meal selection less than older athletes. There were no differences in self-rating across sex, regions and sports, nor stage of competition. There was also no significant relationship between self-rating and previous nutrition advice or label use. There was no correlation between self-rating and any component of the meal. Figure 2 shows examples of photographs and self-ratings for below (5/10) and above (10/10) the median. The comments received on reasons for meal selection (Table 4) were predominately focused on the nutritional attributes (macronutrient content or food content in the meal), sensory factors, sports performance or usual eating practices (food preferences or familiarity).

## 4. Discussion

In this study, we used digital photographs to examine food choices of athletes in a dining hall environment. The purpose was to look at the food variety, diet quality and volume of food selected by athletes and compare this to their reasons for choosing the food and their self-rating of their choices in relation to their sport. Not surprisingly, we found differences between males and females in terms of the volumes of food, energy, carbohydrate, fibre and sodium content of their meals. We also found that more males chose discretionary food and foods from the grains and cereals food group, which were also selected by fewer individuals at lunch. Approximately one third of females did not select any grains for their meal. For those that did choose grains, the number was greater for males than females. This may suggest that the volume of grains and cereals are the differentiating factor between sexes in terms of modifying intake, which directly reflects on intake of carbohydrate. This difference in carbohydrate between sex has been reported previously with the suggestion that females are more likely to restrict their intake to control body weight [25]. We found no relationship in carbohydrate quantity or the choice of grains in the meal to the sporting category, although we have previously shown that athletes from weight category sports self-report eating less grains [3]. We did not differentiate between whole grains and refined varieties, and this may be more critical in determining the quality of the nutrient content of the meal than the total intake.

The majority of athletes chose adequate quantities of all macronutrients, with only one athlete choosing negligible amounts of carbohydrate, and one with minimal protein. The sodium content of meals was high for males in particular (median of 1467 mg) and was correlated to the choice of discretionary food and meats. While this is not an issue for the majority of athletes (in our study, only one athlete identified weight control as a reason for their food choice), for those attempting to make weight, it may be more challenging where recommendations for sodium intake are around 300 mg/MJ [26], and thus, recipe modification to reduce sodium in some meat dishes may be warranted. While assumptions cannot be made about the daily intake of athletes from a single meal selection, it would appear that the energy and macronutrient content of these meals would contribute significantly to a reasonable daily intake for most sports and would also provide adequate quantities both pre- and post-competition. For example, the protein content of the majority of meals was in the range of 20–50 g, which is supportive of a frequent consumption of meals containing protein [27], although we found significantly less protein in the breakfast choices in comparison to other meal periods.

There is currently less evidence around quality of the food athletes are eating. We chose to consider whether meals contained a particular food group as a reflection on whether the athlete intended to eat a variety of foods across different food groups. Both fruit and dairy products were not represented in the majority (approximately 60% and 80%, respectively) of meals. We do not know if individuals chose to not eat these foods or they intended on returning to get them after they had finished. Fruit was readily available as a snack to take outside the dining hall and thus may not have been chosen from the buffet. This may be the case for dairy items such as chocolate milk that may have been used as a recovery drink by some teams. Regardless, the lack of dairy choices in the majority of meals may be of greater concern given that this is a significant source of protein, calcium and vitamin D [20,28] and whey protein, which has been beneficial in lean mass retention during energy restriction [29,30]; however, this did not appear to impact the amount of protein on the plate. Nevertheless, the reasons for the lack of choice of these food groups may be worth exploring in future studies.

A number of athletes commented that they had made food choices that were related to their training and competition. There was a trend to mention macronutrients (particularly carbohydrate and protein) having an influence on their food selection. This is supported by literature that suggests performance factors are important in decisions around food choice in this environment [15,24] but may be less relevant once the competition is over. We found that core food content and fibre was greater in meals prior to competition than post, with a number of athletes mentioning that they were choosing vegetables or ‘greens’.

There was no relationship to use of food labels, and most did not use the label to make their selection. However, we found a higher proportion of females reported using the labelling than males, which is similar to previous studies that found that females place more value on nutrition labelling [10] and use labels more frequently than males [31]. This may be because a larger proportion of the sample had received previous nutrition advice, with most having seen a dietitian. Research in university settings has demonstrated that diet quality may be influenced by the provision of food labels [32,33]; however, this may only be useful in improving dietary intake of athletes when the individual has had less nutrition education. In reality, influence over the food environment and subsequent dietary intake of athletes may be more effective at the level of the caterer [34,35] than food labelling of menu items. Despite this, the provision of labels is necessary for individuals who have specific requirements, such as food allergies or intolerances.

Interestingly, the cohort of participants rated their choices highly despite around one third choosing discretionary foods, the majority not choosing fruit and around one third not choosing vegetables. This may suggest that athletes are not considering the quality of the meals they are selecting but are more focused on the macronutrients that fuel their sport. This is supported by the comments on the reasons for selecting the food, with a large number of participants mentioning carbohydrate and protein. There was a relationship to age, which suggests that younger athletes may feel less confident with the food they chose to eat. Most athletes had previously had nutrition advice, and of these, most had been from a dietitian. Individual rating may be reflective of confidence in their choice, not just a knowledge (or lack) of what would be appropriate for their sport. Alternatively, other factors such as sensory appeal and usual eating practices may take priority over performance, as was suggested in some of the reasons for food selection (Table 4). Interestingly, we found some additional reasons for food selection that differed from those identified in a similar cohort of athletes [24]. These were predominately related to physiological considerations, such as hunger, satiety and gut comfort. However, these factors have recently been identified as important in a cohort of ultra-endurance athletes [14].

The strengths of this study were the lack of interference in the athletes’ food selection, use of photography to capture food selection, the ability to record additional information by the researcher to ensure accurate data collection and concurrent capture of the reasons for food selection, thus not relying on subject recall. However, it is feasible that some foods were missed during data collection and analysis. Sauces, oils, dressings and other condiments may have been overlooked and not recorded in field notes. Furthermore, estimating the serve sizes of these foods is difficult, and this could impact the quantitative analysis in particular. It is also feasible that the self-reported reasons for food selection were influenced by the presence of the researcher. Despite these limitations, this was a relative non-invasive and inexpensive method to collect information about food choice of athletes during competition. Use of computational tools to analyse digital images would be a useful addition as the time taken to code and analyse foods was lengthy.

These results do not indicate actual intake of athletes, as they may not consume all the food that is selected and could return for additional plates of food, nor was this necessarily representative of their intake across the entire day, but this study does give an indication of the types and quality of food that athletes are choosing at major competition events. We did not find differences across geographical regions, but this may be because of the large number of countries that were represented in this study and the limited sample size. As this is the first study to examine meal selection at a large-scale competition event, we aimed to collect data from a wide range of sports and countries to be able to examine the extent of variation in food choice.

Future research could monitor second helpings by athletes as well as photographing waste to get a better understanding of actual dietary intake and to understand the reasons for not consuming certain foods. This would also help caterers to plan the menu and avoid excessive waste, which is increasingly considered a cost and sustainability issue at these major events [36]. It would also be interesting to run this study alongside determinants of food choice [24] and in particular focus on athletes who are making weight or in weight-focused sports where choices from the menu may be more difficult due to restricted offerings.

## 5. Conclusions

This study used a novel method of digital images to determine the food selection of athletes during competition in a large-scale dining facility. The results suggest that athletes may be more focused on quantity of macronutrients rather than quality of their food selection and influenced by a range of factors other than performance, despite having previous nutrition advice. Better education of athletes on diet quality ensuring selection from a range of food groups and improvements to the food environment to assist with food selection could help individuals and teams to plan for the impact of buffet-style eating, where the extensive range of items can lead to inappropriate choices or excessive intake. 

## Figures and Tables

**Figure 1 nutrients-11-01627-f001:**
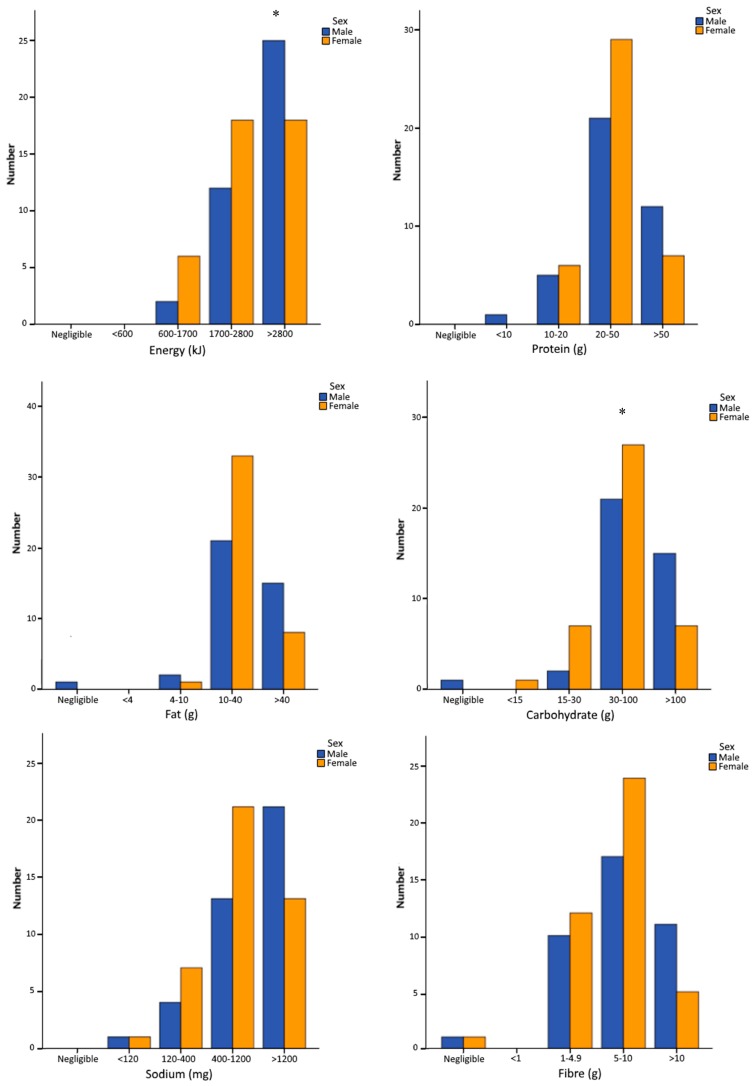
Energy and nutrient content of participants’ meal grouped into categories of negligible, low, medium, high and very high (* Mann–Whitney U test—significant difference between sexes for energy (627.0; *p* < 0.042) and carbohydrate (600.0; *p* = 0.018)).

**Figure 2 nutrients-11-01627-f002:**
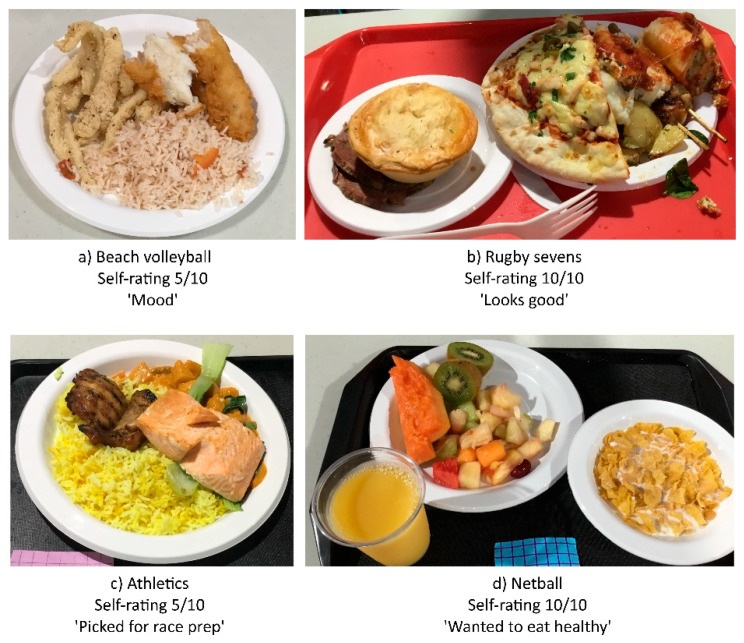
Sample photographs of meals, ratings and reason for choice.

**Table 1 nutrients-11-01627-t001:** Demographic characteristics of participants (*n* = 81).

	Total	Male	Female	Test Statistic; *p*-Value ^^^
Age in years (median, range)	25 (15–60)	27 (17–60)	24.5 (15–39)	NS
Sport category * Weight focused Endurance Power/sprint Racket Team	15 (19.2) 19 (24.4) 12 (15.4) 6 (7.7) 26 (33.3)	6 (16.7) 10 (27.8) 4 (11.1) 4 (11.1) 12 (33.3)	9 (21.4) 9 (21.4) 8 (19.0) 2 (4.48) 14 (33.3)	NS
Region Africa Australia/New Zealand British Isles Canada Caribbean Asia/Pacific	19 (23.5) 18 (22.2) 19 (23.5) 8 (9.9) 5 (6.2) 12 (14.8)	11 (28.2) 8 (20.5) 8 (20.5) 4 (10.3) 2 (5.1) 6 (15.4)	8 (19.0) 10 (23.8) 11 (26.2) 4 (9.5) 3 (7.1) 6 (14.3)	NS
Competition phase Pre/During Post	56 (69.1) 25 (30.9)	25 (64.1) 14 (35.9)	31 (73.8) 11 (26.2)	NS
Experience in dining hall Yes No	31 (38.3) 50 (61.7)	10 (25.6) 29 (74.4)	21 (50.0) 21 (50.0)	5.079 ^c^; 0.021
Previous competition (yes/no) Olympic World Commonwealth Other	23 (37.7) 7 (11.5) 34 (55.7) 22 (36.1)	7 (57.1) 3 (9.1) 18 (54.5) 16 (48.5)	16 (57.1) 4 (14.3) 16 (57.1) 6 (21.4)	8.33 ^c^; 0.004 NS NS 4.81 ^c^; 0.026
Following particular diet ^#^ Performance Preference Allergy/intolerance Macronutrient based	34 (42.0) 10 (29.4) 13 (38.2) 3 (8.8) 8 (23.5)	17 (50.0) 6 (35.3) 8 (47.1) 1 (5.9) 2 (11.8)	17 (50.0) 4 (23.5) 5 (29.4) 2 (11.8) 6 (35.3)	NS
Previous nutrition advice Yes Dietitian Other No	65 (80.2) 59 (90.8) 6 (9.2) 16 (19.8)	29 (74.4) 25 (86.2) 4 (13.8) 10 (25.6)	36 (85.7) 34 (94.4) 2 (5.6) 6 (14.3)	NS
Time last trained or competed Within 24 h Greater than 24 h	59 (72.8) 22 (27.2)	24 (61.5) 15 (38.5)	35 (83.3) 7 (16.7)	4.86 ^c^; 0.025
Use of nutrition label Yes No	30 (37.0) 51 (63.0)	10 (25.6) 29 (74.4)	20 (47.6) 22 (52.4)	4.19 ^c^; 0.041

* Sports categories include: weight-focused (diving, gymnastics, boxing, weight lifting, wrestling); endurance (athletics, cycling and swimming distance events); power/sprint (sprint athletic and swimming events, field events); racket (badminton, table tennis, squash); skill (archery, lawn bowls, shooting); and team (hockey, netball, rugby sevens, basketball, beach volleyball). ^#^ Dietary categories include: performance (making weight, carbohydrate loading, avoiding foods prior to competition /training, adjusting intake for weight, fasting); preference (dislike of foods, cultural familiarity, healthy eating); allergy/intolerance (prawns, nuts, gluten, dairy products, soy, eggs); and macronutrient-based (avoiding macronutrients, macronutrient counting). ^^^ NS = Not significant (*p* > 0.05), tests for differences between groups were with Chi-square test statistic ^c^.

**Table 2 nutrients-11-01627-t002:** Meals tested across meal periods and competition phase.

Meal Period *n* (%)	Total (for Meal Period)	Before/during Competition	After Competition
Breakfast	14 (17.3)	8 (57.1)	6 (42.9)
Lunch	35 (43.2)	27 (77.1)	8 (22.9)
Dinner	32 (39.5)	21 (65.6)	11 (34.4)
Total	81 (100)	56 (69.1)	25 (30.9)

**Table 3 nutrients-11-01627-t003:** Nutrients, food groups, variety and volume of athletes’ meal selection.

**Total (*n* = 81)**		**Sex (*n*, IQR *)**			**Meal Period (*n*, IQR *)**			
**Nutrient**	**Median** **(IQR *)**	**Males**	**Females**	**Test Statistic** ***p*-Value ^^^**	**Breakfast**	**Lunch**	**Dinner**	**Test Statistic** ***p*-Value ^^^**
Energy (kJ)	3025 (2220–3867)	3529 (2457–4825)	2541 (1990–3550)	531.5 ^a^ 0.007	2542 (2209–3529)	3050 (2181–3867)	3212 (2368–4588)	NS
Protein (g)	35 (22–49)	38 (28–58)	31 (21–43)	NS	26 (17–31)	35 (23–52)	42 (26–55)	8.25 ^b^ 0.016
CHO (g)	74 (42–108)	93 (75–122)	49 (32–79)	412.5 ^a^ 0.000	87 (55–106)	50 (33–93)	83 (45–116)	NS
Fat (g)	30 (23–43)	33 (22–50)	28 (24–37)	NS	28 (22–35)	32 (22–41)	31 (25–51)	NS
Fibre (g)	7 (4–10)	9 (4–11)	6 (4–9)	601.0 ^a^ 0.039	5 (3–9)	8 (5–10)	8 (4–11)	NS
Sodium (mg)	1120 (517–1775)	1467 (661–2326)	846 (462–1362)	570 ^a^ 0.019	814 (432–1314)	1238 (662–1654)	1066 (389–1949)	NS
**Number (*n*) of Meals that** **Contained the Food Group (%)**		**Number (n) and Proportion (%) of sex**			**Number (*n*) and Proportion (%) of Meal Period**			
**Food Group**	***n* (%)**	**Males**	**Females**	**Test Statistic; ** ***p*-Value ^^^**	**Breakfast**	**Lunch**	**Dinner**	**Test Statistic; *p*-Value ^^^**
Grains and cereals	63 (77.8)	34 (87.2)	29 (69.0)	3.85 ^c^; 0.05	13(92.9)	22 (62.9)	28 (87.5)	8.10 ^c^; 0.017
Fruit	16 (19.8)	9 (16.7)	7 (23.1)	NS	3 (21.4)	8 (22.9)	5 (15.6)	NS
Fruit juice	12 (14.8)	6 (15.4)	6 (14.3)	NS	3 (21.4)	5 (14.3)	4 (12.5)	NS
Non-starchy vegetables ^#^	56 (69.1)	24 (61.5)	32 (76.2)	NS	0 (0.0)	30 (53.6)	26 (46.4)	38.07 ^c^;<0.0001
Starchy vegetables ^#^	34 (42.0)	13 (33.3)	21 (50.0)	NS	4 (28.6)	14 (40.0)	16 (50.0)	NS
Legumes	7 (8.6)	5 (4.8)	2 (12.8)	NS	2 (14.3)	5 (14.3)	0 (0.0)	NS
Dairy	28 (34.6)	16 (41.0)	12 (28.6)	NS	8 (57.1)	9 (25.7)	11 (34.4)	NS
Meats	68 (84.0)	32 (82.1)	36 (85.7)	NS	7 (50.0)	31(88.6)	30 (93.8)	14.81 ^c^; 0.001
Discretionary	24 (29.6)	18 (46.2)	6 (14.3)	9.85 ^c^; 0.002	7 (50.0)	8 (22.9)	9 (28.1)	NS
**Total (*n*, Range)**		**Sex (*n*, Range)**			**Meal Period (*n*, Range)**			
**Item**	**Median (Range)**	**Males**	**Females**	***p*-Value ^^^**	**Breakfast**	**Lunch**	**Dinner**	***p*-Value ^^^**
Core food groups in meal	3 (1–5)	3 (1–5)	3 (1–5)	NS	3 (2–3)	3 (1–5)	3 (1–5)	6.33 ^b^; 0.042
Bread/cereal serves ^$^ (*n* = 63)	2 (1–2)	2 (0.5–5)	1.5 (0.5–3)	257.5 ^a^; 0.001	2 (1–3)	1.75 (0.5–4)	2 (0.5–5)	NS
Variety (different items)	6 (2–15)	7 (2–15)	6 (3–12)	NS	5 (3–11)	7 (3–15)	7 (2–14)	NS
Volume (plates of food)	1 (1–5)	2 (1–5)	1 (1–3)	592.0 ^a^; 0.016	1 (1–2)	1 (1–4)	2 (1–5)	NS
Discretionary serves ^$^ (*n* = 24)	1.25 (0.25–7.0)	1.5 (0.25–7.0)	1.25 (1–3)	NS	2 (0.5–2.5)	1 (0.25–7)	1.5 (0.5–3.0)	NS

* IQR = Interquartile range. ^#^ Starchy vegetables include potato, corn and peas; non-starchy vegetables include all other varieties. ^^^ NS = not significant (*p* > 0.05); tests for differences between groups were determined by Mann–Whitney U ^a^, Kruskal–Wallis ^b^ or Chi-square test statistic ^c^. ^$^ Based on those that had these food groups present in their meal.

**Table 4 nutrients-11-01627-t004:** Participants’ self-reported reasons for selecting food.

Determinant Related to Food Choice *	Subcategory	*n*	Sample Comments
Nutritional attributes	Macronutrient content	18	*Felt like some carbohydrate, sticking to protein-based meals.* *High carb.* *Protein. Fibre. Carbohydrate.* *Bit of protein. Bit of carbohydrate (but not simple) and greens.* *Wanted carbohydrate and protein and flavour.*
	Food groups/food content	11	*Love tuna. Some type of protein, vegetables and carbohydrate always at lunch.* *Go for greens meat and vegetables.* *Always cheese protein and carbohydrate at lunch and greens at dinner.*
Sensory	Taste	15	*Likes the taste.* *What’s available and what would taste good.* *Taste, try to keep a variety of healthy foods.*
	Appearance	12	*Looked colourful.* *Looked good.*
Performance	Competition and training	20	*About to train.* *Heading to a game soon so needed for before it.* *Picked for race prep.* *Carbohydrate for race tomorrow.* *Water content of the fruit –wanted to hydrate the body.*
Usual eating practices	Preferences	13	*Chose based on what he liked.* *Nothing else wanted in Western breakfast.* *Whatever I feel like.*
	Familiarity	8	*Familiar in foreign country.* *Repetitive breakfast. Knows it safe.* *…. Corn fritters mum makes so reminded her of home.* *What you usually eat.*
	Exploratory eating	2	*Wanted to try different things.* *Done racing – trying different foods.*
	Nutrition plan	1	*Following the diet she is supposed to, and foods she likes.*
Food/health awareness		3	*Wanted to eat healthy. Standard diet.* *…, try to keep a variety of healthy foods. Don’t avoid anything.* *Read up nutrition advice himself. YouTube, men’s health website.*
Emotional influences		3	*Mood.* *How I’m feeling.*
Weight control		1	*Trying to maintain weight.*
Influence of others		1	*Team mate said corn fritters were really good and likes bacon and eggs.*
**Other factors** ^#^			
Physiological reasons	Gut comfort	4	*Light meal.* *Simple. ….. Had it before so won’t be bad on stomach.* *Wanted something light - competing in 3 h.*
	Hunger	2	*Was hungry after training.* *How hungry I am. ….*
	Satiety	2	*…. Keeps him full.*
Others	Weather/climate	1	*Feeling the heat here so wanted something light and refreshing.*
	Availability	1	*What’s available and what would taste good.*
	Health/health condition	1	*Food restriction and intolerances.*

***** Categories based on athlete food choice questionnaire (AFCQ) [24]. ^#^ Additional factors not part of the AFCQ.

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
