# Peer review of "Evaluation of Athletes’ Food Choices during Competition with Use of Digital Images"

_nutrients, 2019, doi:10.3390/nu11071627_

Round 1

Reviewer 1 Report

In this manuscript, the authors investigate the food intake of college athletes by taking pictures of the participants’ meals in a dining hall. They find that many athletes choose rather unhealthy meals that do not contain fruit or dairy. Unhealthy choices, however, are not accurately reflected in the participants’ evaluation of their meal. While I believe that this topic is of interest to the journal’s audience, some issues should be addressed to increase the clarity of the manuscript.

Major comments:

1) The title might need to be amended to “Assessing food choices of athletes during competition with use of digital images” or changed to a title that provides more information about the findings.

2) Abstract: Please add some information on the study’s background.

3) General remark: In the introduction (p.2), the authors state that the dining hall environment during competitions is a special eating environment. While I generally agree with this statement, I would suggest to move this part to the discussion and discuss the special environment with regard to future investigations and limitations of the present study, because the authors did not compare the food choices in the dining hall/ during a competition with eating occasions in the participants’ usual environment. Given that many participants listed the event as a reason for their selection, this suggests that their food choice during a competition might differ from food choices made at home/ when not competing.

4) General remark: In addition, only one meal was assessed. Studies show that there is large intrapersonal variation in food choice (e.g. König & Renner, 2018), thus it might be important to take into account how typical the meal is, both for eating occasions during competitions and in general, when evaluating the healthiness and drawing conclusions about the need for interventions. As I suppose that this was not assessed in the present study, the authors might need to add this as a limitation.

Reference: König, L. M., & Renner, B. (2018). Colourful= healthy? Exploring meal colour variety and its relation to food consumption. Food Quality and Preference, 64, 66-71.

5) Materials and methods: The data collection procedure should be described in a separate paragraph and not in the paragraph on participants.

6) Materials and methods: Could the fact that participants had to report their answers to a research assistant have influenced the results, especially the reporting of reasons? E.g. due to impression management, more participants might have reported to have chosen the meal due to its macronutrients/ being a nutritious meal during a competition.

7) Materials and methods: Why were reasons for food choice assessed with an open ended question and not using a standardized comprehensive questionnaire? Were participants able to list more than one reason?

8) Materials and methods: Was the self-rating of the food selection framed so that it related to health, or was it a general evaluation? This might have impacted what attributed participants took into account and therefore might also impact how closely one would expect this item to be related to the objectively determined healthiness.

9) Materials and methods: Why is the evaluation of total served, variety and volume listed under qualitative assessment? This should be moved to quantitative assessment.

10) Materials and methods: Please provide indicators for interrater agreement and accuracy of the estimates of energy and macronutrient content.

11) Materials and methods: Do “discretionary” food items equal stereotypical unhealthy foods? Please give examples.

12) Materials and methods: How was the sample size determined? Given that only one meal was assessed per participant and for some analyses, the meal type is taken into account which further reduces the number of meals analysed, this is a rather small sample compared to other picture-based studies, including McClung et al. (2017) that is referenced in the manuscript. This should be added as a limitation.

13) Materials and methods: Please provide more information on the statistical tests conducted, including full test-statistics and effect sizes.

14) Table 1: The authors might need to refrain from conducting statistical analyses involving region, as some groups are very small. Instead, authors might need to think about forming larger groups.

15) Results: The authors might need to consider removing the heading 3.1 as this is the only subheading in this section.

16) Results: Results for aim 2 were hard to find as they are hidden in the description of Figure 1. Please move the results to the main text.

17) Results: In the abstract, authors compare the self-rating to the objective evaluation of dietary healthiness. Is this based on descriptive analysis or were any tests conducted. Please provide more information on this comparison, also taking into account comment 8.

Minor comments:

18) Was the food self-served or plated by a member of staff?

19) The authors should check the manuscript for punctuation errors, e.g. l. 72, l. 86.

Author Response

Thank you for your comments. We have responded below. 

Review 1 response:

In this manuscript, the authors investigate the food intake of college athletes by taking pictures of the participants’ meals in a dining hall. They find that many athletes choose rather unhealthy meals that do not contain fruit or dairy. Unhealthy choices, however, are not accurately reflected in the participants’ evaluation of their meal. While I believe that this topic is of interest to the journal’s audience, some issues should be addressed to increase the clarity of the manuscript.

Major comments:

1)     The title might need to be amended to “Assessing food choices of athletes during competition with use of digital images” or changed to a title that provides more information about the findings.

We have changed the title to “Evaluation of athletes’ food choices during competition with use of digital images”

2)     Abstract: Please add some information on the study’s background.

The abstract has been revised to include ‘The selection of foods made by athletes during competition can impact on performance, yet to date the quality of their food choices has not been investigated.’(lines 9-10)

General remark: In the introduction (p.2), the authors state that the dining hall environment during competitions is a special eating environment. While I generally agree with this statement, I would suggest to move this part to the discussion and discuss the special environment with regard to future investigations and limitations of the present study, because the authors did not compare the food choices in the dining hall/ during a competition with eating occasions in the participants’ usual environment. Given that many participants listed the event as a reason for their selection, this suggests that their food choice during a competition might differ from food choices made at home/ when not competing.

Thank you for your feedback and yes, it's correct that this is likely to differ. However, the purpose of this study was not to compare choices between different environments. We feel it is important to introduce the reader to this environment in the introduction to provide context.  The unique environment of the competition dining hall has previously been established in past studies and this has been shown to result in inadequate nutrient intake of athletes. A reference to support this statement has been added to the introduction in line 46.  In addition, the following sentence in the discussion has been revised as follows:

‘These results do not indicate the actual intake of athletes, as they may not consume all the food that is selected and could return for additional plates of food, nor was this necessarily representative of their intake across the entire day, but this study does give an indication of the types and quality of food that athletes are choosing at major competition events’ (lines 314-317)

 General remark: In addition, only one meal was assessed. Studies show that there is large intrapersonal variation in food choice (e.g. König & Renner, 2018), thus it might be important to take into account how typical the meal is, both for eating occasions during competitions and in general, when evaluating the healthiness and drawing conclusions about the need for interventions. As I suppose that this was not assessed in the present study, the authors might need to add this as a limitation. Reference: König, L. M., & Renner, B. (2018). Colourful= healthy? Exploring meal colour variety and its relation to food consumption. Food Quality and Preference, 64, 66-71.

This has been added to the limitations in lines 314-317 as per comment the above.

3)     Materials and methods: The data collection procedure should be described in a separate paragraph and not in the paragraph on participants.

This has been revised so that it is in a separate paragraph in lines 87-97

4)     Materials and methods: Could the fact that participants had to report their answers to a research assistant have influenced the results, especially the reporting of reasons? E.g. due to impression management, more participants might have reported to have chosen the meal due to its macronutrients/ being a nutritious meal during a competition.

Yes, this could have influenced their reporting of the reasons for food selection. For this reason, this has been mentioned as a limitation to the study in the discussion as follows: ‘It is also feasible that the self-reported reasons for food selection were influenced by the presence of the researcher.’ (lines 309 to 310)

5)     Materials and methods: Why were reasons for food choice assessed with an open-ended question and not using a standardized comprehensive questionnaire? Were participants able to list more than one reason?

At the time of the study, no validated tool for food choice in athletes existed. We have made reference to a recent publication of a new tool in our discussion (line 326). While this would have been interesting to examine, this would have not provided responses specific to the individual meal. Food choice questionnaires are valid for a broader perspective on determinants influencing food choice and not so much about a specific meal selection. For example, the responses around choosing food specifically related to the macronutrients in their meal would not have been captured with this tool. For this reason, open-ended responses were more appropriate in this study. 

6)     Materials and methods: Was the self-rating of the food selection framed so that it related to health, or was it a general evaluation? This might have impacted what attributed participants took into account and therefore might also impact how closely one would expect this item to be related to the objectively determined healthiness.

The self-rating of food selection was framed in terms of a performance nutrition context (ie. in relation to their sport). This has been added to the text in line 95 of methods. 

7)     Materials and methods: Why is the evaluation of total served, variety and volume listed under qualitative assessment? This should be moved to quantitative assessment.

The two headings have been consolidated into one single subheading – data analysis - see line 98

8)     Materials and methods: Please provide indicators for interrater agreement and accuracy of the estimates of energy and macronutrient content.

A more detailed explanation of the coding and the inter-rater agreement has been added to the methods (lines 103-106 and 123 to 130). Any variation in coding is relevant to the determination of serving sizes of items and food groups on the plate. This was ultimately used to calculate the energy and macronutrient analysis which was previously established for each menu item based on 100g and per serve.  

9)     Materials and methods: Do “discretionary” food items equal stereotypical unhealthy foods? Please give examples.

Discretionary foods are those that recommended to be consumed in limited quantities according to the AGHE and are generally considered unhealthy.  Further clarification and examples have been added to the text in lines 114-116.   

Materials and methods: How was the sample size determined? Given that only one meal was assessed per participant and for some analyses, the meal type is taken into account which further reduces the number of meals analysed, this is a rather small sample compared to other picture-based studies, including McClung et al. (2017) that is referenced in the manuscript. This should be added as a limitation.

We recognise that this is a small sample, however, this was not a validation study as per McClung or Christoph and is unique to the competition environment where access to athletes can be extremely limited.  Recognition of the small sample size and image of a single meal has been acknowledged in the limitations. See lines 314-321

10)  Materials and methods: Please provide more information on the statistical tests conducted, including full test-statistics and effect sizes.

The test statistics have been added to tables, figures, and text where relevant.

11)  Table 1: The authors might need to refrain from conducting statistical analyses involving region, as some groups are very small. Instead, authors might need to think about forming larger groups.

These regions have been used previously with international cohorts of athletes (see Burkhart, S.J. and F.E. Pelly, IJSNEM, 2013. 23(1): p. 11-23). Larger groups would not provide the information that would be relevant to eating styles. We feel that this should be left as is to be consistent with other studies even though this was not statistically significant.

12)  Results: The authors might need to consider removing the heading 3.1 as this is the only subheading in this section.

An additional heading ‘3.1 Participant characteristics’ has been added (line 158)

13)  Results: Results for aim 2 were hard to find as they are hidden in the description of Figure 1. Please move the results to the main text.

Apologies – this text should have been included in the results, not under the Figure. This has been corrected in lines 203-212.

14) Results: In the abstract, authors compare the self-rating to the objective evaluation of dietary healthiness. Is this based on descriptive analysis or were any tests conducted. Please provide more information on this comparison, also taking into account comment 8.

The results of the self-rated analysis have been provided in lines 203-212. This was accidentally omitted from the results and added to Figure 1 as per comment 13 above. Self-rating of food intake was tested against various demographic characteristics and dietary analysis as outlined in this additional text. We have not included all of these results in the abstract due to the word limit as it was not the primary aim of the study.

Minor comments:

15) Was the food self-served or plated by a member of staff?

Some items were self-serve and some served by staff. When served by staff, they were instructed to serve to the size listed on the nutrition label. The athlete could request more serves of the item as desired.

16) The authors should check the manuscript for punctuation errors, e.g. l. 72, l. 86.

Thank you – this has been rectified.

Reviewer 2 Report

Dear authors,

It was a pleasure to read your paper; however, there are some questions that I would like to hear your responses. 

Have you noticed any significant differences between athletes in the team sports with individual in any dependent variables?  I am asking this as to whether there was any noticeable food culture in the  team sports.

Given that 90 percent of the participated athletes received nutrition advice from Dietitians,  I am wondering how education can help with food choices (referring to line 309-310). It would be useful if you could provide more detailed here as practical application specifically focusing on  a buffet style dining hall settings. 

Author Response

Thank you for your comments. We have responded below. 

Question: Have you noticed any significant differences between athletes in the team sports with individual in any dependent variables?  I am asking this as to whether there was any noticeable food culture in the team sports.

We didn't test team sports versus other sports as there were only 26 participants. We did test weight category versus non-weight category sports but found no significant differences. A larger sample would be needed to detect any particular differences across sporting categories. 

Question: Given that 90 percent of the participated athletes received nutrition advice from Dietitians,  I am wondering how education can help with food choices (referring to line 309-310). It would be useful if you could provide more detailed here as practical application specifically focusing on a buffet style dining hall settings. 

We have added a comment about educating athletes about diet quality and selecting from a range of food groups in this environment. See lines 331- 335. It will be important to better to understand the reasons for food selection prior to undertaking education strategies. (see lines 323 - 326)

Round 2

Reviewer 1 Report

The revision has already greatly improved the manuscript. However, there are some remaining concerns that need to be resolved:

1) The authors still do not provide effect sizes for all statistical tests, and p-values are omitted if they are not < .05. To be able to interpret the results given the small sample size, and to allow for the inclusion of this study in meta-analyses, the authors should provide both effect sizes and all p-values.

2) As there are many questionnaires for determinants of food choice are already available that could have been adapted before the study, reason for using open-ended format for assessing reasons for food choice should be provided in the manuscript.

3) Please check references in text: Sometimes, a space is missing between the word and the bracket.

Author Response

Thank you for your response. We have addressed your comments below.

Comment 1:  We have considered the necessity to include non-significant results and effect sizes for all tests for possible inclusion in any future meta-analysis.  However, as this study is focused on the quality of the food selected as opposed to dietary intake, it would not be appropriate to include this study in a meta-analysis. This was a sample of convenience hence the reasons for the smaller sample size. As stated in the manuscript; 'As this is the first study to examine meal selection at a large-scale competition event, we aimed to collect data from a wide range of sports and countries to be able to examine the extent of variation in food choice (lines 322-324)'. This data represents a single meal from diverse individuals in a unique environment. Additionally, this would affect the eligibility of inclusion in future meta-analyses. We feel the statistical tests and p values of non-significant results distract from the significant findings, hence why they have not been included. 

Comment  2: We have added the following sentence to the manuscript; Open-ended responses to individual reasons for food choice were preferred over existing survey tools as the purpose of the study was to investigate specific reasons for the meal selection, not broader factors influencing food choice. (lines 94-96)

Comment 3:  This has been fixed. 

Reviewer 2 Report

I have no further comments for this manuscript.

Author Response

Thank you for your feedback